# The impact of skim reading and navigation when reading hyperlinks on the web

**Gemma Fitzsimmons**[1]*, **Lewis T. Jayes**[2], **Mark J. Weal**[3], **Denis Drieghe**[1]

**1** School of Psychology, University of Southampton, Southampton, United Kingdom, **2** School of Psychology, University of Surrey, Guildford, United Kingdom, **3** School of Electronics and Computer Science, University of Southampton, Southampton, United Kingdom

☯ These authors contributed equally to this work.
* gemma.fitzsimmons@soton.ac.uk

**Data Availability Statement:** The data underlying the results presented in the experiments in this manuscript are available from the UK Data Service. The DOI is: 10.5255/UKDA-SN-854153.

## Abstract

It has been shown that readers spend a great deal of time skim reading on the Web and that this type of reading can affect lexical processing of words. Across two experiments, we utilised eye tracking methodology to explore how hyperlinks and navigating webpages affect reading behaviour. In Experiment 1, participants read static Webpages either for comprehension or whilst skim reading, while in Experiment 2, participants additionally read through a navigable Web environment. Embedded target words were either hyperlinks or not and were either high-frequency or low-frequency words. Results from Experiment 1 show that while readers lexically process both linked and unlinked words when reading for comprehension, readers only fully lexically process linked words when skim reading, as was evidenced by a frequency effect that was absent for the unlinked words. They did fully lexically process both linked and unlinked words when reading for comprehension. In Experiment 2, which allowed for navigating, readers only fully lexically processed linked words compared to unlinked words, regardless of whether they were skim reading or reading for comprehension. We suggest that readers engage in an efficient reading strategy where they attempt to minimise comprehension loss while maintaining a high reading speed. Readers use hyperlinks as markers to suggest important information and use them to navigate through the text in an efficient and effective way. The task of reading on the Web causes readers to lexically process words in a markedly different way from typical reading experiments.

## Introduction

When we investigate the real-world task of reading online text (termed hypertext [1]), we need to take into consideration the way reading on the Web differs from the reading for comprehension task that is traditionally used in reading research. Typically, reading research uses trials that contain a single, stand-alone sentence (for a review, see [2]) to explore lexical processing. The reader is asked to read the sentence for comprehension and there will often be a comprehension question following the sentence. If the participant has a low accuracy for answering these usually rather easy questions, the experimenter knows that the participant was

**Funding:** GF was funded by an EPSRC grant for the Doctoral Training Centre in Web Science: EP/G036926/1. This work formed a part of a PhD completed in the Web Science DTC. The funders had no role in study design, data collection and analysis, decision to publish, or preparation of the manuscript.

**Competing interests:** The authors have declared that no competing interests exist.

not fully engaged with the task. This process is of course different from everyday reading. Additionally, while many experiments have examined reading paragraphs, in the vast majority of reading experiments, participants only read a single line of text and as such, do not have to integrate information across multiple sentences. As useful as this experimental design is for exploring factors that affect lexical processing within a controlled setting, this is not a common reading behaviour that people engage in in everyday life. Reading on the Web is clearly a very different task. Primarily, there is so much information on the Web it is less likely that anyone will be able to read all available information for comprehension due to time constraints. Moreover, often not all text is equally important to the reader and/or their task. To make the task of reading on the Web more manageable the reader may skim read to try and gain as much important information as possible in the most efficient way. Skim reading is a commonplace strategy whereby readers employ some form of rapid, selective reading strategy, often omitting words [3–5]. Another difference with typical lab-based reading experiments is the presence of hyperlinks. Hyperlinks refer to words that enable users to navigate from one webpage to another, when clicked. hyperlinks can have a two-fold impact on readers. Firstly, hyperlinks are salient words in the text and can signal to the reader where important information may lie in the text. Secondly, hyperlinks serve as a tool for navigation between Webpages. A decision needs to be made about which hyperlinks to click on to navigate to other Webpages.

In this paper, we conducted two experiments exploring task differences between reading for comprehension versus skim reading as well as the impact of hyperlinks, by specifically looking at how these tasks and text elements affect lexical processing. Experiment 1 displayed Web pages, but the user could not click or navigate. In Experiment 2 a functional, clickable environment was utilised, where the impact of clicking and navigating on reading and comprehension of the text was investigated. The combination of these two experiments allows us to dissociate the unique impact of hyperlinks as a salient text item that potentially indicates important information from the impact caused by the hyperlink being a navigational tool as well. Unlike previous research, the current studies investigate the effect of navigation and skim reading on lexical access during reading. The use of eye tracking allows for a temporally sensitive measure of the degree of processing readers engage in during reading words on the Web, and how navigation and skim reading affect this, providing novel insight into the depth of processing of words during the real-world task of reading online text.

## Skim reading

When reading outside of the laboratory, people may 'skim' through text and not fully process all aspects of the text that has been presented to them. Research literature suggests that reading on the Web is more likely to involve skim reading [3, 4]. Liu suggests that screen-based reading behaviour is characterised by 'more time spent browsing and scanning, keyword spotting, one-time reading, non-linear reading, and reading more selectively, while less time is spent on in-depth reading, and concentrated reading' [3, pp.700].

Previous research has directly compared reading for comprehension and skim reading. In an early experiment, Just and Carpenter explored skim reading and compared eye movements to when the participants were engaged in reading for comprehension [6, 7]. They found that skim readers were about two and a half times faster than normal readers. Furthermore, their eye movement analysis showed that the skim readers fixated on fewer words than normal readers. When examining gaze durations on words, Just and Carpenter also found them to be shorter for skim readers, who spent on average 100ms less time on each fixation (around one–third of the average fixation time during normal reading) [6]. However, even with this reduction in fixation times the skim readers still showed effects of word frequency (low frequency

words had longer fixation times compared to high frequency words, [6–8]) and word length (longer words had longer fixation times compared to shorter words [9–12]) This was similar to those seen in normal readers, but the sizes of these two effects were much smaller. Clearly, eye movement behaviour is affected by the task of skim reading versus reading for comprehension.

There is a great deal of evidence suggesting that during skim reading some comprehension is lost [6, 13–17]. One of the causes for this loss in comprehension could be that readers can often solve comprehension problems by re-reading the text that has caused the issue. There is, however, very little re-reading when skim reading, perhaps due to the self-imposed time constraints that are caused by the nature of skim reading. However, loss in comprehension is not consistent across all of the text being read. There appears to be a difference between information regarded as important or unimportant. The important information does not receive the same loss of comprehension that is observed for the unimportant information [5, 15, 18]. To explain these findings, it has been suggested that the reader engages in an adaptive strategy in order to gain as much information from the text as possible, in a reduced time.

## Clicking, navigating and decision making

One of the main differences between reading plain text and hypertext is the fact that hypertext is non-linear and has no strict route through the information. Conklin suggests that a reader could easily become 'lost in hyperspace' when trying to navigate through a website due to the mass interlinking of webpages [19]. McKnight, Dillon and Richardson suggested that the unknown scope of the hypertext document could lead to incorrect assumptions about the scope of the documents' content and result in a poor reading strategy [20]. In linear text it is much easier to see the scope of the document and browse through the content. Dillon, Richardson and McKnight argue that if the user does not know how the information is organised, it makes it more difficult to find specific information [21]. In comparison, paper-based documents, such as books, tend to have a convention for how the information is organised, such as index pages and contents pages, which catalogue the location of topics and convey the overall organisation of the whole text.

It is not just the issue of getting lost in a hypertext environment that we need to consider; there is also the issue of the large amount of choices and decisions that need to be made. Elm and Woods suggest that users may be overwhelmed and disorientated by the sheer amount of choice offered by a complex, large network of information [22]. The users may not understand the structure of the system and what potentially exists in the hypertext document. McDonald and Stevenson argue that although a large linear text can also be confusing for a reader, there are typically a number of discourse cues such as page numbers, contents listings and headings that the reader can take advantage of [23]. Non-linear hypertext lacks a lot of these types of cues. The same text presented as a linear document may cause no issues to the reader, but in its hypertext format, it might lead to navigational problems where the reader is confused by the non-linear structure.

There is an on-going debate about whether in-text hyperlinks hinder reading. Carr suggests that hyperlinks within the text are a distraction and hinder comprehension of the text. He argues that having to evaluate hyperlinks and navigating a path through them is a demanding task that substantially increases readers' cognitive load and thereby weakens their ability to comprehend and retain what they are reading ([24], pp.126).

Carr's argument is based on research investigating the cognitive load of hypertext on users [e.g. 25], which suggests that comprehension increased when participants read plain text compared to when they read hyperlinked text. However, their study is somewhat limited in being

able to generalise to other forms of hypertext, including reading webpages. The text used by Miall and Dobson [25] was a piece of literary fiction that had been converted to hypertext and hyperlinks were added to it. The text had not originally been created to be displayed in a hypertext format, making the hyperlinked document quite artificial. This artificial hypertext document may be the reason for the increase in cognitive load, in turn making it difficult to generalise these results to reading on the Web. This being said, other research does corroborate with Carr's suggestion that extra cognitive demands are associated with having to make decisions about whether to follow hyperlinks [24].

Some researchers have explored working memory and the concept of cognitive load and its impact on reading hypertext. DeStefano and LeFevre conducted a review of cognitive load in relation to reading hypertext [26]. They argued that the extra task demands of reading hypertext causes an increased cognitive load to the readers in comparison to linear text. Because the readers must make decisions about which hyperlinks to follow, additional cognitive demands are placed on working memory. Recently, Scharinger, Kammerer and Gerjets measured both the EEG and pupil size of readers engaging in a task that closely simulated hypertext reading and link selection [27]. They found evidence of increased load on executive functions when the reader had to perform hyperlink-like selection.

It is not just the decision of whether or not to click a hyperlink that could increase cognitive load. The reader's decision to follow a hyperlink and explore different content could interrupt on-going comprehension processes. Comprehension involves the creation and development of situation models, which are complex mental representations that the reader instantiates in order to integrate statements from the text they are reading into their knowledge [28]. For example, Dee-Lucas and Larkin found that hyperlinks in text distract users by interrupting information processing [29]. While reading, users may stop to click on hyperlinks in the middle of text content, thus interrupting their cognitive processing and leaving the reader with a fragmented representation of the text content. Because of the nonlinear nature of hypertext, when a reader is reading text on one topic on a webpage, if they choose to click a hyperlink, it takes them to another webpage. This new webpage may contain content that is unrelated to the content they have just come from on the previous webpage. This could cause disruption to the reader's development of a situation model and result in the readers' comprehension of the text being reduced.

These suggestions of disruption caused by hyperlinks were questioned in a previous set of experiments [30]. Most importantly, in an environment resembling a Wikipedia page, we demonstrated that, at least when reading for comprehension, the use of blue hyperlinks does not have a negative influence on reading. This was shown by a lack of an effect of a word being a hyperlink compared to when it was not a hyperlink on eye movement behaviour (specifically, early fixation measures or skipping probability). However, one single effect did demonstrate that readers treat hyperlinked words differently to non-hyperlinked text; readers were more likely to re-read text when encountering a low frequency, hyperlinked word [30]. Specifically, when encountering a low-frequency word, we observed an increase in re-reading of the preceding text but more so when the word was also a hyperlink compared to when it was not. This finding suggests that hyperlinks highlight important information and suggest additional content which, for more difficult concepts, invites rereading of the preceding text. However, while interesting, the task used in this study was not entirely typical of reading on the Web, as readers read for comprehension only, and on static, experimentally manipulated Wikipedia pages. The experiments reported here aimed to build upon this finding by testing a more typical environment for reading on the Web. In Experiment 1, this is done through the introduction of the task of skim reading, to explore the impact of hyperlinks within this task on both eye movements and comprehension within a static Webpage environment. In Experiment 2,

readers are also allowed to navigate through their environment, by clicking links, providing an environment matching typical reading on the Web.

Through the addition of these task elements, we have explored whether the impact of hyperlinks on lexical processing differed from our original findings when participants are engaged in a reading behavior more typical of Web browsing. Furthermore, by exploring skimming and navigation separately, we could dissociate the effects of these two manipulations on reading within a controlled environment. While Experiment 1 is not typical of reading on the Web, due to the lack of navigation, it does allow for a baseline to be set of how skim reading affects lexical processing of text within static Webpages. This allows us to draw comparisons with Experiment 2 to disentangle the unique effects of reading for comprehension and skim reading when navigating through a dynamic Web environment.

Our predictions regarding the degree of lexical processing are made on the basis that the frequency effect is traditionally considered a reliable indices of lexical processing. The frequency effect is a robust effect within psycholinguistic literature [6–8], and has previously been used to assess skim reading [4, 5, 29]. Furthermore, it has been taken as an indices of the depth of lexical processing during tasks such as visual searches of word lists (e.g. reduced depth of lexical processing compared to reading for comprehension as indicated by a lack of word frequency effect [31]) and proofreading (e.g. larger frequency effect when proofreading due to increased levels of lexical processing [32]). Following in this tradition, we investigated depth of lexical processing using the frequency effect. From previous research we predicted that readers would read faster when asked to skim read, but would have reduced comprehension [6]. We predicted that we would observe shorter fixation times and more word skipping in the skim reading condition. However, we also predicted that because the linked target words are salient, they might attract the attention of the reader in the skim reading condition resulting in less skipping of linked words. We also included a word frequency manipulation in this experiment in order to explore whether common lexical effects are present in hyperlinked text and to investigate if they are modulated by the word being hyperlinked. Our previous research [30] suggested that, if reading behaviour is unchanged by introducing skim reading and the ability to click links, we will observe a frequency effect, whereby low frequency words are less likely to be skipped and have longer fixation time. Whether the target word is linked or not, however, will modulate the effect in re-reading, such that low-frequency hyperlinks elicit more re-reading.

## Experiment One

### Method

**Participants.**   Thirty-two native English speakers (2 male, 30 female) with an average age of 20.00 years (range– 18–31) participated in exchange for course credits or payment (£9) and were members of the University of Southampton community. All had normal or corrected-to-normal vision and no known reading disabilities. None of the participants took part in Experiment 2. All samples reported in this paper are typical of eye movement and reading studies.

**Apparatus.**   Eye movements were measured with an SR-Research Eyelink 1000 eye tracker operating at 1000 Hz (1 sample every millisecond). Participants viewed the stimuli binocularly, but only the right eye was tracked. Words were presented in 14pt mono-spaced Courier font. The participant's eye was 73 cm from the display; at this distance three characters equalled about 1˚ of visual angle.

**Materials and design.**   The stimuli in Experiment 1 consisted of forty edited Wikipedia articles (example stimuli available: https://goo.gl/JLvvMD) taken from Experiment Three of Fitzsimmons et al. [30]. One-hundred and sixty target words were embedded in sentences

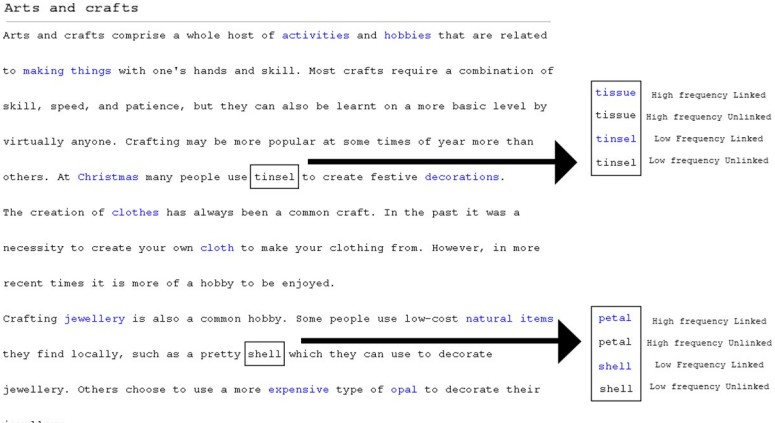

**Fig 1. Example Wikipedia stimulus with examples of high and low frequency words in linked and unlinked form.** Note. Wikipedia branding removed from example for copyright purposes–full version of stimuli can be seen here: https://goo.gl/JLvvMD.

(one target word per sentence) and four sentences were inserted into each Wikipedia article. The text was created by taking existing Wikipedia articles on neutral topics and inserting four experimental sentences into the existing text. The experimental sentences were designed to be semantically consistent with the text already present, so as not to stand out from the existing text. The rest of the text on each screen was identical to the source material on Wikipedia, including additional words that were linked, for additional naturalness. This decision was made so that the articles were as close to a natural Web environment as possible, while gaining the additional control experimental sentences. The Wikipedia articles were ten to twelve lines long. The target words were nouns and the location of the target words were scattered across the sentences, but they were never on the start or end of a line. All these design decisions were made to align with the traditional eye movements and reading methodology. The target words within these articles were either displayed in blue or black to denote if the word was a hyperlink or not, respectively (see Fig 1).

In total there were 8 conditions in a 2 (Task Type: Comprehension, Skimming) x 2 (Word Type: Linked, Unlinked) x 2 (Word Frequency: High, Low) within participants design. At a target word level, the target words within these articles were either displayed in blue or black to denote if the word was a hyperlink or not. There was also a word frequency manipulation where the frequency of the target word was either high or low frequency. The word frequencies were taken from the Hyperspace Analogue to Language (HAL) corpus [33]. The frequency norms were used to extract both high and low frequency words to create the experimental stimuli. The high frequency words had an average log transformed HAL frequency of 9.94 and the low frequency words had an average log transformed HAL frequency of 5.81. There was a significant difference between the high and low word frequency stimuli, $t(159) = 29.66$, $p <$ .001. All target words were 4–7 characters in length with an average of 5.60 characters and the high/low frequency pairs were matched on word length. The various versions of each stimulus were presented according to a Latin square design, meaning every participant saw only one version of each edited Wikipedia article.

**Procedure.** Before any of the experiments in this article took place, ethics approval was applied for, peer-reviewed and granted by the University of Southampton Psychology Department Ethics Committee. Ethics approval was sought and approved for all experiments within this article. Participants were given an information sheet and a verbal description of the

experimental procedure and informed that they would be reading passages on a monitor while their eyes were being tracked. The text on the screen gave the instructions to read either for comprehension or to skim read. This was blocked such that the first twenty stimuli were to be read for comprehension and the second twenty to be skim read.

When the skim reading portion of the experiment began the participants were instructed to 'skim read as you would naturally, as if you are reading a large text book that you need to read quickly'. Participants were told there was no time limit, and they simply had to skim read naturally. We did not counterbalance the Task Type. Participants were not told they were going to be skim reading until just before that half of the experiment was due to begin, so as not to influence the first part of the experiment which was to be read for comprehension. We worried if participants were first asked to skim read, it may become difficult to slow down and read "normally". This was also suggested by participants in a pilot study. Participants during piloting of the study indicated they found it much easier to adapt to skim reading after typical reading behaviour than vice versa. Both experiments took 60–90 minutes, with breaks, which are not atypical of other eye movement and reading studies, ensuring tiredness had a reduced influence on our results.

The participants' head was stabilised in a head/chin rest to reduce head movements that could adversely affect the quality of the calibration of the eye tracker. At the beginning of each trial the participant had to look at a fixation point on the screen. When the eye tracker registered a stable fixation on the fixation point, the stimulus was displayed ensuring that the first fixation fell at the beginning of the text. When participants finished reading they confirmed they had finished by pressing a button on the response box in front of them.

The participants were informed that they were to respond to comprehension questions presented after each trial when four comprehension questions were presented to the participants, one at a time. The comprehension questions were designed to be simple and only ever required a yes or no response. They asked about information across the whole webpage (not just the target sentences), ensuring readers were reading the entire passages of text. Participants responded to the questions by pressing the appropriate button on a response box. They were designed to ensure readers were reading the text and understanding the text, as such, they ensured task validity in our reading task. After the questions the next trial would appear.

## Results

Trials where there was tracking loss were removed prior to the analysis. Fixations shorter than 80ms that were within one character of the previous or following fixation were merged and all fixations shorter than 80ms or longer than 800ms were removed to eliminate outliers, resulting in the removal of 5.43% of the total dataset [33, see also: 34]. Finally, when calculating the eye movement measures, data that were more than 2.5 standard deviations from the mean for a participant within a specific condition were removed (<1% of dataset). Data loss affected all conditions similarly.

For the local target word analyses an interest area was drawn around each target word. The interest area is the size of the target word including the space preceding it. The local analyses below are conducted using the fixations that landed on the target word, within the interest area drawn around it.

We focused our analysis on three key eye movement measures: Skipping probability, single fixation duration and go-past times (means shown in Table 1). Skipping probability is the probability that a target word does not receive a direct fixation during the first-pass. Single fixation duration is the duration of the fixation if the reader made exactly one first-pass fixation on the target word. Go-past time is the time between first fixating the word and moving past it

**Table 1. Means of eye movement measures for Experiment 1.** Standard deviation in parentheses.

| Task Type | Word Type/ Word Frequency | Skipping Probability (%) | Single Fixation Duration (ms) | Go-past time (ms) |
|---|---|---|---|---|
| Comprehension | Linked/High | 52 (20) | 221 (44) | 378 (223) |
| | Linked/Low | 48 (22) | 233 (37) | 370 (164) |
| | Unlinked/High | 54 (19) | 212 (37) | 322 (116) |
| | Unlinked/Low | 51 (20) | 246 (45) | 375 (140 |
| Skimming | Linked/High | 52 (23) | 201 (27) | 295 (128) |
| | Linked/Low | 48 (22) | 221 (35) | 284 (76) |
| | Unlinked/High | 68 (18) | 204 (41) | 263 (94) |
| | Unlinked/Low | 63 (18) | 205 (31) | 250 (50) |

to the right (including any time fixating previous content via regressions that originate from the target word). In this experiment when the target word was fixated, in 93.91% of the cases it received a single fixation. Therefore, we limited the fixation duration analyses to when there was a single fixation on the target word.

We ran Linear Mixed Models (LMMs) using the lme4 package (Version 1.1–12) in R [35] to explore the impact of three variables. Logistic General Linear Mixed Models were used for the skipping probability measure. The three independent variables were included as fixed factors: Task Type (Comprehension, Skimming), Word Type (Linked, Unlinked) and Word Frequency (High, Low). Participants and items were included as random effects variables. A maximal random model was initially specified for the random factors [36]. If a model did not converge, the random effect structure was pruned first by removing the interactions between the slopes, then correlations in the random structure and finally by successively removing the slopes for the random effects explaining the least variance until the maximal converging model was identified. Additionally, the interactions between Word Frequency and Task Type, Word Frequency and Word Type, and the three-way interaction between Word Frequency, Word Type and Task Type were removed from the skipping probability model, as comparisons showed they did not contribute towards the fit of the model. The go-past time model excluded the interaction between Word Frequency and Word Type, and the three-way interaction between Word Frequency, Word Type and Task Type for the same reason. All the patterns observed in the models were identical whether they were run on log-transformed or untransformed fixation durations, allowing us to present the data run on the untransformed fixation durations in order to increase transparency. The only exception was for go-past times measures where the fixation times were log transformed. This was due to the data needing to be normalised because it was skewed and resulted in qualitatively different models for log transformed versus untransformed go-past times. All fixed effects estimates are shown in Table 2 and were calculated using successive differences contrasts so that the intercept corresponds to the grand mean. Absolute values of *t* equal to or bigger than 1.96 were interpreted as significant because for high degrees of freedom as is typically the case in LMMs, the *t* statistic approximates the *z* statistic.

**Comprehension.** The comprehension question accuracy was consistently high, regardless of whether the reader was reading for comprehension of skim reading, with 89% of questions correctly answered (Reading for comprehension: 91%; Skim reading: 86%).

**Word skipping.** There was a main effect of Word Frequency in skipping probability such that the high frequency words were skipped significantly more often than the low frequency words (see Table 1). There was also a main effect of Task Type, where there was more skipping when the reader was skim reading compared to when they were reading for comprehension.

**Table 2. Fixed effect estimates for skipping probability percentage of the target word and the fixation times on the target word in ms for Experiment 1.**

| | Skipping Probability | | | Single Fixation Duration (ms) | | | Go Past Time (ms) | | |
|---|---|---|---|---|---|---|---|---|---|
| | Estimate | Std. Error | z value | Estimate | Std. Error | t value | Estimate | Std. Error | t value |
| Intercept | 0.21 | 0.13 | 1.66 | 219.77 | 3.91 | 56.25 | 5.60 | 0.03 | 164.89 |
| Word Frequency | -0.19 | 0.06 | -3.05 | 16.03 | 3.15 | 5.09 | 0.06 | 0.03 | 2.02 |
| Word Type | 0.43 | 0.06 | 7.07 | -1.47 | 4.30 | -0.34 | -0.05 | 0.03 | -1.70 |
| Task Type | 0.30 | 0.06 | 4.94 | -16.62 | 3.22 | -5.16 | -0.19 | 0.02 | -9.62 |
| Word Type x Task Type | 0.57 | 0.12 | 4.74 | 5.62 | 7.10 | 0.79 | -0.05 | 0.04 | 1.28 |
| Word Frequency x Task Type | | | | -17.66 | 6.27 | -2.81 | -0.12 | 0.04 | -3.07 |
| Word Frequency x Word Type | | | | 14.64 | 6.35 | 2.31 | | | |
| Word Frequency x Word Type x Task Type | | | | 33.19 | 12.61 | 2.63 | | | |

Random structure for skipping: (1 + Frequency|Participant) + (1|Item); Random structure for single fixation duration: (0 + Frequency+ Word Type|Participant) + (1|Items); Random structure for go-past reading time: (1 + Frequency * Word Type|Participant) + (1|item)

This replicates the research conducted by Just and Carpenter [6] who found similar results in skim reading.

In skipping probability, as well as the significant main effects of Word Frequency and Task Type, there was also a main effect of Word Type. When the target word was unlinked it was more likely to be skipped compared to when it is linked. This effect of Word Type was qualified by an interaction with Task Type (see Fig 2). Subsequent contrasts showed that there was no difference in skipping probability when the target word was linked or unlinked during comprehension reading ($z = 1.46$, SE = 0.09, $p = .150$), but there was a significant difference in the skim reading condition. Linked words were significantly less likely to be skipped compared to unlinked words in the skim reading condition ($z = 7.54$, SE = 0.09, $p < .001$). This suggests that when the readers are skim reading they are attempting to fixate the linked words more than the unlinked words and avoid skipping them.

**Fixation duration measures.** There was a main effect of Word Frequency in single fixation duration. The low frequency words had significantly longer fixation durations than the high frequency words. This replicates previous research where low frequency words are fixated for longer because they are more difficult to process than highly frequent words [9]. There was also a main effect of Task Type in single fixation durations where there were shorter fixation

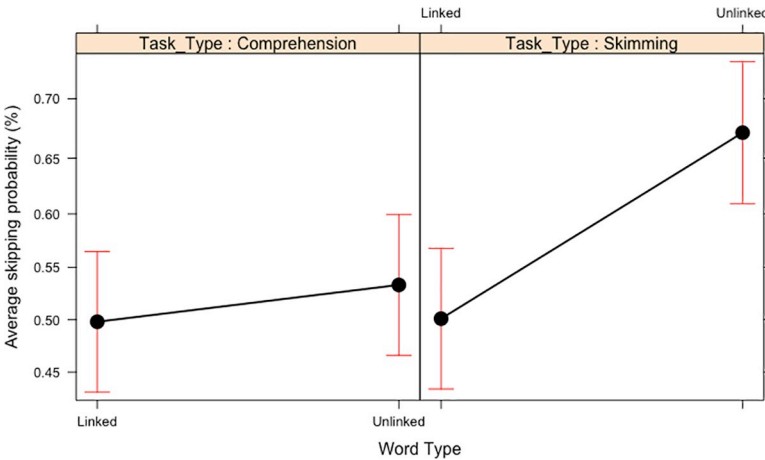

**Fig 2. Two-way interaction between Word Frequency and Task Type in Experiment 1.** Means and standard error bars for skipping probability.

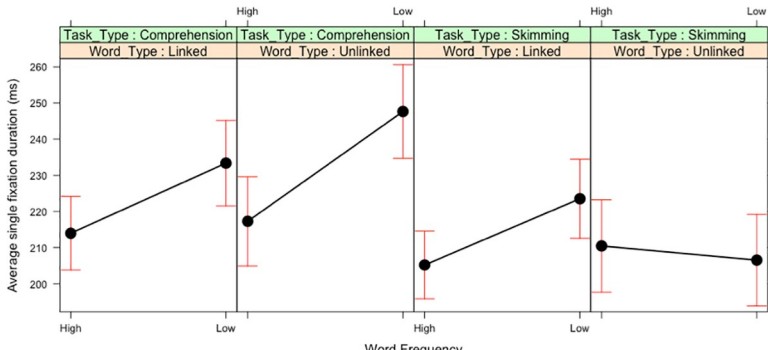

**Fig 3. Three-way interaction between Word Frequency, Word Type and Task Type in Experiment 1.** Means and standard error bars for single fixation durations.

durations when the participant was skim reading. This replicates the research conducted by Just and Carpenter [6] who found similar results in skim reading.

For single fixation duration, these main effects were qualified by multiple two-way interactions, and these were qualified by a three-way interaction between Word Frequency, Word Type and Task Type (see Fig 3). To explore this three-way interaction, additional contrasts were conducted.

As Fig 3 clearly shows, the three-way interaction was caused by the fact that, when participants skim read the passages, a frequency effect only emerged for linked target words (b = 8.71, SE = 3.77, $t$ = 3.03), and not for unlinked target words (b = -2.12, SE = 3.37, $t$ = -0.63). In contrast, when participants read for comprehension, there was a significant frequency effect both for linked target words (b = 9.48, SE = 3.23, $t$ = 2.94) and unlinked target words (b = 14.80, SE = 3.32, $t$ = 4.46).

Although the majority of fixations on the target word were single fixations, when the target word was fixated 14.11% of target words had regressions to previous interest areas. We also explore go-past times, however, we need to point out that the re-reading analysis will not have a high amount of statistical power, as re-reading was obviously rather rare.

*Go-past times*. In go-past times the main effects of Word Frequency and Task Type were qualified by a two-way interaction (see Fig 4) whereby the frequency effect is present when

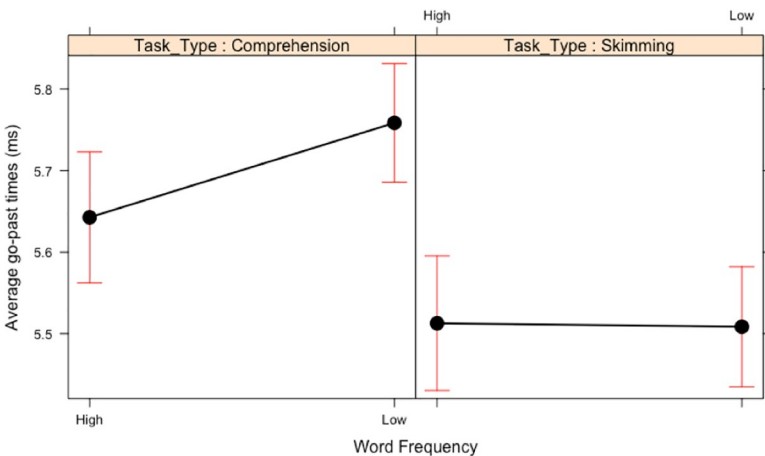

**Fig 4. Two-way interaction between word frequency and Task Type in Experiment 1.** Means and standard error bars for log-transformed go-past time.

**Table 3. Means and Standard deviations for trial duration in Experiment 1.** Standard deviation in parentheses.

| Task Type | Trial Duration (seconds) |
|---|---|
| Comprehension | 31.12 (7.44) |
| Skimming | 15.86 (3.97) |

reading for comprehension (b = 0.13, SE = 0.03, $t$ = 4.60), but is missing in skim reading (b = -0.01, SE = 0.03, $t$ = -0.46).

Finally, we measured trial duration (See Table 3). This is a global measure, so we only included the independent variable of Task Type (Comprehension, Skimming) as a fixed factor. The intercept was allowed to vary for the items and participants variable. We found a main effect of Task Type where the trial duration (See Table 4) was significantly longer when it was read for comprehension compared to when it was skim read, validating our manipulation of skim reading.

## Discussion

Experiment 1 demonstrated that skim reading has a pronounced influence on lexical processing. We observed that fixations were shorter and skipping rates were higher, on average, for the skim reading condition compared to the reading for comprehension condition, replicating the findings of Just and Carpenter [6]. However, the most interesting finding here is in relation to the impact that hyperlinks have on reading behaviour. Fitzsimmons et al. [30] found that, when reading for comprehension, hyperlinks had only a limited impact upon reading behaviour, restricted to increased re-reading of the low frequency, hyperlinked words. We replicated the results of Fitzsimmons et al., finding that hyperlinks are not a hindrance to reading when reading for comprehension in early eye movement measures, but we did not observe increased re-reading induced by a low-frequency hyperlink, although re-reading was rare in this experiment.

When the reader was skim reading, however, we observed several notable differences in eye movement behaviour compared to reading for comprehension. Firstly, readers were less likely to skip hyperlinks when skim reading. Secondly, during skim reading, we found a frequency effect for single fixation duration for hyperlinked words, but not for unlinked words. When taken as a proxy for depth of lexical processing, this lack of frequency effect clearly indicates that unlinked words are not as fully processed when readers are engaged in skim reading. This effect of skim reading reducing lexical processing, critically, is not present in the case of hyperlinks.

Regardless of whether a reader skim reads or reads for comprehension, the frequency effect was observed for hyperlinks. This suggests that even when reading quickly, hyperlinks are still important signals to the reader that they should be fully processed in order to engage in an efficient skimming pattern. This finding suggests that readers could be engaging in a speed-comprehension trade-off that is optimal for the task at hand. Participants may have wanted to read quickly while still retaining as much comprehension as possible. They may have been using the links as anchor points or signals throughout the text if the links denote the most important

**Table 4. Fixed effect estimates for trial duration in seconds for Experiment 1.**

| | Estimate | Std. Error | $t$ value |
|---|---|---|---|
| Intercept | 31435.32 | 849.70 | 37.00 |
| Task Type | -15303.82 | 105.63 | 144.88 |

information. Typographical signals have been shown to result in the reader paying more attention to the signalled content [37] and often in improved memory for the signalled text [37–40]. It has also been previously shown that hyperlinks can assist in helping learners retain information [41], perhaps because they are working as a typographical signal. From these findings, we suggest that participants used the hyperlinks as markers for the presence of important information and used them in a strategy to skim read through the text in the most efficient way possible.

## Experiment Two

Hyperlinks are visually salient and important navigational tools during reading of hypertext, as they represent a link to other content on the Web. Experiment 1 further displayed the importance of hyperlinks in signalling a unit of important text within a passage. This supports previous evidence showing hyperlinks highlight important information, thus affecting skim reading behaviour [30, 42]. However, we also need to consider how the reader reads the text when it contains clickable hyperlinks (i.e. when reading hypertext).

In Experiment 1, the reader could not click and navigate the environment. While a clear limitation of Experiment 1, as the readers were not strictly reading hypertext, this was implemented to maintain experimental control by simplifying the experience as much as possible in order to explore the impact of hyperlinks during skim reading, without yet introducing the added complexity of navigation and clicking. In Experiment 2, we run a similar manipulation to that seen in Experiment 1, where the task was manipulated (reading for comprehension or skim reading). The target words within the text were also manipulated to be either high or low word frequency and were displayed either in blue (linked) or black (unlinked). Additionally, in Experiment 2 we allow the links to clicked, which also means that if the page was re-visited, the link would be made purple as if the links have been visited, consistent with how they would normally look on the Web. The reader chooses the next trial by clicking on the hyperlink they wish to go to, simulating a realistic Web environment. We predicted that we will find mostly the same effects as in Experiment 1. However, by allowing the reader to navigate the links we might expect to observe inflated fixation durations for the linked words in total reading times where the reader may spend longer on the linked words to evaluate which link to click to navigate to another page. As such, Experiment 2 clearly builds upon Experiment 1 by having the novel inclusion of navigation through hypertext, allowing us to investigate the unique effect of navigating on lexical processing.

### Method

**Participants.**   Thirty-two native English speakers (15 male, 17 female) with an average age of 20.03 (range– 18–34) years participated in exchange for payment (£9) and were members of the University of Southampton community. None of the participants had taken part in Experiment 1. All had normal or corrected-to-normal vision and no known reading disabilities.

**Apparatus.**   The apparatus was identical to the one used in Experiment 1, with the addition of a computer mouse that participants could use to click links using the cursor on-screen.

**Materials and design.**   Experiment 2 was similar in design to Experiment 1. The forty wiki pages used in Experiment 1 were insufficient to allow for a realistic Web environment with full clicking and navigable functions. As such, the stimuli for Experiment 2 consisted of 843 edited new Wikipedia articles with experimental sentences inserted into the existing text. The Wikipedia articles were two to twelve lines long (example stimuli available:http://bit.ly/2wLk0QQ). Participants could follow any hyperlinks they wished to click on and because of this environment, the number of target words observed by each participant varied dependent on the pages they choose to view. The Wikipedia pages contained between zero and four target words. All

target words were 4–7 characters in length with an average of 5.12 characters and the high/low frequency pairs were matched on word length. The high frequency words had an average log transformed HAL frequency of 9.62 and the low frequency words had an average log transformed HAL frequency of 6.02 [according to the norms collected in the HAL corpus, 43]. There was a significant difference in frequency between the high and low word frequency stimuli, $t(471) = 49.24$, $p < .001$.

The experiment took place as four sessions with different starting pages. The first two sessions were read for comprehension and the last two sessions were skim read. The participant was instructed to read for comprehension or to skim read the text (dependent on the session) and to then choose any hyperlink they wished to follow. If the participant wished they could also go back to as many pages as they wished with a designated back button. Participants could use this back button at any time to re-read a page, follow a different link on a previous page or simply to navigate away from a page that was a dead-end containing no hyperlinks in the text to click. The participant was told they may occasionally have to answer a comprehension question, which could be about any part of the text displayed, after reading some of the articles. Due to the large structure of this experiment and the fact that participants could choose any hyperlinks they wished to follow, we constrained the experiment by ending each session after the participant had visited ten unique pages in each session, totalling forty unique pages per participant [for more detail of the development of the stimuli, please see 44].

In total there were 8 conditions in a 2 (Task Type: Normal, Skimming) x 2 (Word Type: Linked, Unlinked) x 2 (Word Frequency: High, Low) within participants' design. The participants were instructed to read the text on the screen either for comprehension or to be skim read. As in Experiment 1, we did not counterbalance the Task Type because the normal reading blocks may have been influenced by first having to skim read and being informed of the skim reading condition may have influenced their normal reading behaviour. At a target word level, the target words within these articles were either displayed in blue or black to denote if the word was a hyperlink or not respectively (and would be displayed in purple if that link had been visited). The display was 73 cm from the participant's eye and at this distance three characters equal about 1° of visual angle.

**Procedure.** The procedure was the same as Experiment 1, except to move onto the next trial the participants needed to choose a hyperlink to click on to navigate to the next trial topic. Participants were informed that they were going to navigate sections of Wikipedia. Each participant viewed four starting sections, which we will refer to as blocks. For each block, they were told they would view a page, read it as normal, and select a link based on what they wanted to read about next. Participants were instructed to read the whole page to a satisfactory degree to be able to answer a comprehension question. Participants were not informed of the number of pages they would read. After the participant encountered ten unique articles the experimenter would set up the next block with a new starting page. Each participant viewed at least 40 unique articles over the four blocks. The first two blocks were read for comprehension, the second two blocks were skim read. The participant was not informed about the skim reading blocks until just before they began, so as not to influence the reading of the first two block.

When participants finished reading the page they were on, they selected a link within the text that they wished to follow or pressed a button on the keyboard that they were told corresponds to the "back button" on a browser which would go back to the page they previously visited. Participants could go back as many pages as they wished and could click any hyperlink of the page they were on. Comprehension questions were presented to the participants if that page had a comprehension question attached to it, on average participants were presented with comprehension questions on 45% of trials. The comprehension questions were related to the text on the article that had just been read, were simple and required a true or false response.

The comprehension questions were presented to ensure the participants were reading and comprehending the text displayed to them and to measure the level of comprehension across the tasks. Participants responded to the questions by pressing the appropriate response on the screen with the mouse cursor. The appropriate next page that had been selected by the participant would then subsequently appear. If the participant had visited a page with a comprehension question before, the question was not displayed again. The experiment lasted approximately 90 minutes and participants could take a break between trials whenever they required.

## Results

**Comprehension accuracy.** The comprehension question accuracy was 62% (Reading for comprehension: 66%; Skim reading: 60%). Comparing these scores to Experiment 1 (Reading for comprehension: 91%; Skim reading: 86%), the current experiment has much lower accuracy scores. This could be interpreted in one of two ways. Either the act of navigating has a serious impact on comprehension or another difference between the current experiment and Experiment 1 has caused this reduction in comprehension. For example, in Experiment 1 there were four comprehension questions shown after each article. For the current experiment, however, there was only one comprehension question shown after 45% of trials, on average. The reader may have given more attention to the task of reading and navigating and found the comprehension questions an unusual aside to the actual task. This is considered further in the general discussion.

**Eye movement analyses.** The data were cleaned before the eye tracking analysis and this process was identical to Experiment 1 (resulting in the removal of 5.89% of the total dataset). When calculating the eye movement measures, data that were more than 2.5 standard deviations from the mean for a participant within a specific condition were removed (<1% of dataset). Data loss affected all conditions similarly. Additionally, due to the design of the experiment, whereby readers navigate their way through the stimuli set, it was possible for readers to not fixate or skip all targets within a single trial. Target words were only included in the analysis if readers fixated on the sentence where a target word appeared. All eye movement measures for Experiment 2 are listed in Table 5.

In Experiment 1 when the target word was fixated, in 93.91% of the cases it received only a single fixation. Therefore, for analysing Experiment 1 we limited the fixation duration analysis to when there was a single fixation on the target word. In the current experiment, however, the target word was fixated only once 54.95% of the time. Multiple fixations on target words were clearly more likely, compared to Experiment 1, due to the additional navigation and decision-making aspects of Experiment 2. Due to this fact we also include a wider range of fixation

**Table 5. Means and Standard Deviations Experiment 2.** Standard deviation in parentheses.

| Task Type | Word Type/Word Frequency | Skipping Probability (%) | First Fixation Duration (ms) | Single Fixation Duration (ms | Gaze Duration (ms) | Go-Past Time (ms) | Total Reading Time (ms) |
|---|---|---|---|---|---|---|---|
| Comprehension | Linked/High | 12 (32) | 197 (54) | 201 (54) | 273 (137) | 1065 (4215) | 542(389) |
| | Linked/Low | 15 (36) | 212 (64) | 224 (67) | 305 (169) | 1728 (4954) | 572 (370) |
| | Unlinked/High | 19 (39) | 207 (66) | 211 (64) | 256 (111) | 322 (341) | 311 (179) |
| | Unlinked/Low | 18 (39) | 215 (64) | 219 (64) | 275 (128) | 443 (1320) | 322 (192) |
| Skimming | Linked/High | 20 (40) | 205 (63) | 207 (67) | 240 (105) | 388 (970) | 437 (310) |
| | Linked/Low | 14 (34) | 228 (68) | 239 (65) | 293 (120) | 461 (938) | 501 (343) |
| | Unlinked/High | 19 (39) | 202 (60) | 204 (56) | 227 (87) | 245 (147) | 245 (110) |
| | Unlinked/Low | 15 (36) | 204 (67) | 200 (61) | 217 (89) | 247 (284) | 229 (109) |

measures to explore the data, compared to those examined in Experiment 1, where they would not provide additional insight. These include first fixation duration, gaze duration and total reading time. First fixation duration is the duration of only the first fixation on a target word; gaze duration is the summed duration of fixations from the first fixation on the target word until readers made a saccade away from the target word; total reading time is the summed fixation of all fixations made within a region during a trial.

Total reading time is traditionally used as it incorporates all re-reading that might occur, as it sums all fixations on a word within a trial together. However, in the current experiment, when the participants are choosing which hyperlinks to click, they may re-read these links, but not for traditional reasons to re-read the text such as problems with integrating the meaning of words and sentences. In the current experiment, the reader may re-read links in order to select and decide which links to click to navigate away from that article to a new article. Therefore, the total time on the target word not only represents re-reading, a measure of processing difficulty associated with the specific word, but also the additional time to make the decision to click a hyperlink or not. It is very difficult to pull apart the re-reading time and the decision time in the eye movement measures, but we can assume any inflated times observed for the linked target words could be due to either re-reading and time devoted to processing the high-level information that content is linked to the target word [30] and/or the decision making related to clicking (unique to the current Experiment).

A series of linear mixed-effects models (LMMs) using R (Version 1.1–12 - [35]) were used to examine the eye movement measures which explored the impact of three variables (binominal General Linear Mixed models were used for the skipping probability measure). These three independent variables were included as fixed factors: Task Type (Comprehension, Skimming), Word Type (Linked, Unlinked) and Word Frequency (High, Low). Participants and items were included as random effects variables. Random effect structures were constructed in the same manner as Experiment 1. All eye movement measures were log transformed, in order to normalise skewed data. As log transformed data provided qualitatively different models to untransformed data, we therefore present the results of the log transformed data here. The three-way interaction between word type, task type and frequency were removed from the gaze duration model because it did not contribute towards the fit of the model. The means are presented in Table 5 and the output of the LMM's in Table 6.

**Word skipping.** For skipping probability, there was no effect of any of the fixed factors. Experiment 1 showed increased word skipping of unlinked words, especially of unlinked words when skim reading, yet we did not replicate this effect in the current experiment. On average, there was less skipping in this experiment than in Experiment 1, which could have caused the differences in the findings (Average skipping probability: Current experiment: 17%; Experiment 1: 55%).

**Fixation duration measures.** For first fixation duration and single fixation duration, there was a main effect of Word Frequency where low frequency words received longer fixation durations than high frequency words. Additionally, there was an interaction between Word Type and Task Type. Specifically, when reading for comprehension, we observed no difference in fixation times for Word Type (first fixation duration: $b = -0.03$, SE = 0.03, $t = -1.25$; single fixation duration: $b = -0.02$, SE = 0.04, $t = -0.63$), but when skim reading, the unlinked words received significantly shorter fixation durations than linked words (first fixation duration: $b = 0.06$, SE = 0.03, $t = 2.29$; single fixation duration: $b = 0.08$, SE = 0.03, $t = 2.68$—see Fig 5). This suggested that when skim reading, unlinked words were judged as less important and, therefore, the reader spent less time reading them.

For gaze duration, there was a main effect of Word Frequency, where low frequency words received longer gaze durations. There was a main effect of Task Type, where skim reading

**Table 6. Fixed effect estimates for skipping probability percentage of the target word and the fixation times on the target word in ms for Experiment 2.**

| | Skipping Probability | | | First Fixation Duration (ms) | | | Single Fixation Duration (ms) | | |
|---|---|---|---|---|---|---|---|---|---|
| | Estimate | Std. Error | z value | Estimate | Std. Error | t value | Estimate | Std. Error | t value |
| Intercept | -2.01 | 0.17 | -12.17 | 5.30 | 0.02 | 304.52 | 5.32 | 0.02 | 278.02 |
| Word Frequency | -0.08 | 0.19 | -0.39 | 0.06 | 0.02 | 3.07 | 0.08 | 0.03 | 2.59 |
| Word Type | 0.27 | 0.19 | 1.37 | -0.02 | 0.02 | -0.91 | -0.04 | 0.03 | -1.61 |
| Task Type | 0.10 | 0.18 | 0.57 | 0.00 | 0.02 | -0.17 | -0.02 | 0.03 | -0.60 |
| Word Frequency x Word Type | 0.19 | 0.50 | 0.38 | -0.05 | 0.06 | -0.84 | -0.12 | 0.07 | -1.66 |
| Word Frequency x Task Type | -0.53 | 0.35 | -1.52 | 0.01 | 0.04 | 0.16 | -0.01 | 0.05 | -0.16 |
| Word Type x Task Type | -0.41 | 0.35 | -1.17 | -0.10 | 0.04 | -2.73 | -0.11 | 0.05 | -2.16 |
| Word Frequency x Word Type x Task Type | 0.68 | 0.70 | 0.97 | -0.04 | 0.07 | -0.58 | -0.10 | 0.10 | -1.01 |

| | Gaze Duration (ms) | | | Go Past Time (ms) | | | Total Reading Time (ms) | | |
|---|---|---|---|---|---|---|---|---|---|
| | Estimate | Std. Error | z value | Estimate | Std. Error | t value | Estimate | Std. Error | t value |
| Intercept | 5.48 | 0.02 | 240.42 | 5.62 | 0.04 | 142.56 | 5.76 | 0.03 | 172.34 |
| Word Frequency | 0.12 | 0.03 | 4.16 | 0.19 | 0.06 | 2.95 | 0.08 | 0.04 | 1.91 |
| Word Type | -0.11 | 0.03 | -4.06 | -0.29 | 0.07 | -4.42 | -0.49 | 0.04 | -11.08 |
| Task Type | -0.11 | 0.02 | -4.59 | -0.25 | 0.06 | -4.32 | -0.21 | 0.04 | -5.84 |
| Word Frequency x Word Type | -0.18 | 0.08 | 2.18 | -0.31 | 0.15 | -2.09 | -0.23 | 0.12 | -1.89 |
| Word Frequency x Task Type | 0.00 | 0.05 | 0.04 | -0.08 | 0.10 | -0.79 | -0.04 | 0.07 | -0.53 |
| Word Type x Task Type | -0.11 | 0.05 | 2.24 | 0.09 | 0.10 | 0.87 | -0.08 | 0.07 | -1.17 |
| Word Frequency X Word Type x Task Type | | | | -0.09 | 0.23 | -0.37 | -0.08 | 0.14 | -0.58 |

Random structure for skipping: (1|Participant) + (1|Item); Random structure for first fixation duration: (0 + Frequency + Task Type|Participant) + (1|Item); Random structure for single fixation duration: (0 + Frequency + Word Type + Task Type|Participant); Random structure for gaze duration: (1 + Frequency + Word Type * Task Type|Participant) + (1|Item); Random structure for go-past reading time: (1 + Frequency + Word Type + Task Type|Participant) + (1|Item); Random structure for total reading time: (0+ Frequency * Word Type + Task Type|Participant)

resulted in shorter gaze durations. There was also a main effect of Word Type, where unlinked words received shorter gaze durations. These main effects were qualified by a two-way interaction between Word Type and Task Type (see Fig 5), which was similar to the two-way interaction we observed in first fixation duration and single fixation duration where there is no significant effect of Word Type when reading for comprehension ($b = 0.04$, SE = 0.04, $t = 0.93$), but there was an effect of Word Type when skim reading. The unlinked words received shorter gaze durations when skim reading ($b = 0.15$, SE = 0.04, $t = 3.92$), compared to

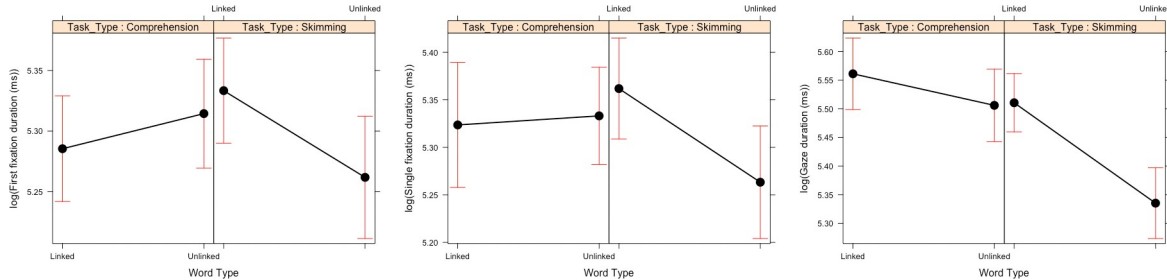

**Fig 5. Two-way interaction between Word Type and Task Type for Experiment 2.** Means and standard error bars for first fixation duration, single fixation duration and gaze duration.

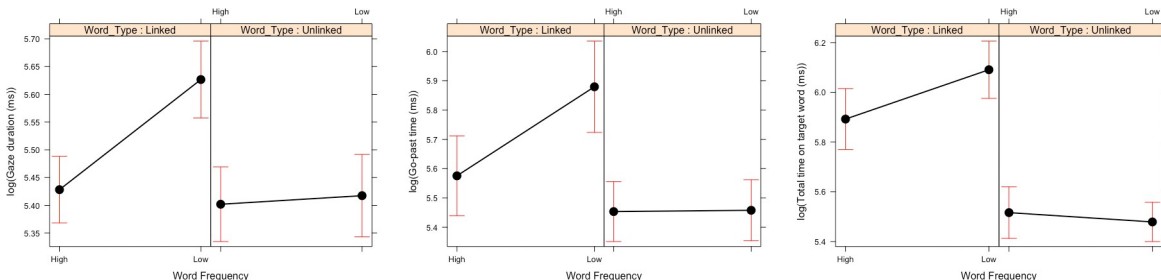

**Fig 6. Two-way interaction between Word Frequency and Word Type for Experiment 2.** Means and standard error bars for gaze duration, go-past time and total time.

linked words. This indicates that when skim reading, readers were more likely to skim over the unlinked words, resulting in shorter first pass reading times, compared to the linked words.

For gaze duration, there was also a two-way interaction between Word Frequency and Word Type (see Fig 6). There was a frequency effect for linked words ($b$ = -0.21, SE = 0.05, $t$ = -4.20), with low frequency words receiving longer gaze durations. This frequency effect, however, was not present for unlinked words ($b$ = -0.06, SE = 0.05, $t$ = -1.14), suggesting that the unlinked target words were not lexically processed to the same degree as the linked target words, resulting in shorter first pass reading times.

For go past times, there was a main effect of Task Type, where there were shorter go-past times during skim reading compared to reading for comprehension. There was also a main effect of Word Frequency, where low frequency words yielded longer go-past times than high frequency words and a main effect of Word Type where the unlinked words received shorter go-past times compared to the linked words. These latter 2 main effects were qualified by a two-way interaction between Word Frequency and Word Type (see Fig 6). A frequency effect is present when the word is linked ($b$ = -0.25, SE = 0.10, $t$ = -2.39), with low frequency words receiving longer go-past times. However, the frequency effect was not present for the unlinked words ($b$ = 0.02, SE = 0.07, $t$ = 0.29). Again, this suggested that when the target word is unlinked it was not being fully lexically processed.

For total reading time on the target word, there was a main effect of Task Type, where the target words received shorter total fixations in the skim reading condition compared to reading for comprehension. There was also a main effect of Word Type, where the unlinked words received shorter total fixations compared to hyperlinks. And there was a marginal main effect of Word Frequency, with a suggestion of low frequency words receiving longer total fixation times than high frequency words. These effects were qualified by a marginal two-way interaction between Word Frequency and Word Type (see Fig 6), again suggesting a frequency effect present when the word is linked ($b$ = -0.53, SE = 0.09, $t$ = -5.62), with low frequency words receiving longer total fixations. This frequency effect was not present for the unlinked words ($b$ = -0.06, SE = 0.07, $t$ = -0.87). Again, this suggested that when the target word is unlinked it was not being fully lexically processed. This is the two-way interaction between Word Frequency and Word Type that was also present in gaze duration and go-past times.

We have already mentioned that in this experimental scenario, total reading time on the target word interest area, in all likelihood, also represents the decision time when deciding which link to click to navigate to the next page. We see inflated total fixation times for the linked words compared to the unlinked word (513 ms on average for linked words and 277 ms on average for unlinked words). These differences are much larger than those seen for gaze duration, suggesting these differences are due to revisits to the linked words after first past reading. This indicates that readers were spending much more time on the linked words after

**Table 7. Means and Standard deviations for trial duration in Experiment 2.** Standard deviation in parentheses.

| Task Type | Trial Duration (seconds) |
|---|---|
| Comprehension | 39.04 (14.98) |
| Skimming | 21.97 (7.88) |

first-pass reading. This was because linked target words were being revisited, as part of the decision-making process, in order to choose which link to navigate and click on.

Finally, we measured trial duration (See Table 7). This is a global measure, so we only included the independent variable of Task Type (Comprehension, Skimming) as a fixed factor. The intercept was allowed to vary for the items variable. For the participants' variable the intercept, as well as the slope for Task Type was allowed to vary. We found a main effect of Task Type where the trial duration (See Table 8) was significantly longer when it was read for comprehension compared to when it was skim read.

## Discussion

In comparison to Experiment 1, in Experiment 2, readers had the ability to freely navigate the hypertext environment. Readers read the text presented to them in edited Wikipedia pages and could then click on links they wanted to navigate to. This is a novel experiment in that it is one of the first to explore eye movements and reading in a Web-like environment that is well-controlled, allows for clicking, and with eye movements recorded with millisecond temporal accuracy. Experiment 2 is also very important in that it attempts to replicate the previous work in Experiment 1 and in Fitzsimmons et al. [30], but in a 'real' Web environment, where the reader can click and navigate as well as read. Consistent with the results of Experiment 1, we once again found there was a reduction in the time spent on unlinked compared to linked words when skim reading (see Fig 5) for early measures of reading (first and single fixation and gaze duration). This is consistent with the important finding that links receive deeper levels of lexical processing during skim reading.

There were several differences with Experiment 1. Firstly, we did observe overall lower skipping rates compared to Experiment 1 and no significant effects for this measure. In all likelihood, this reduction in skipping was a consequence of our requirement for readers to start reading a sentence in order to include skipping of or landing on a target word within that sentence into our analysis. Readers could end a trial in Experiment 2 without reading the entire passage, by clicking on a link. We could not, therefore, include target words that featured in unread sentences in our skipping analysis, because this represents a biproduct of being able to navigate, rather than part of the lexical processing of skim reading/reading for comprehension. This makes direct comparisons of the two experiments for skipping measures difficult.

Another key difference from Experiment 1 emerged in our later measures of gaze duration and go-past reading time measures, where we again observed a frequency effect, but only for linked words (this was also a marginal interaction for total reading time–see Fig 6). We did not observe the modulation of task on this interaction from Experiment 1, whereby the frequency effect only occurred for linked words when skim reading but occurred for linked and

**Table 8. Fixed effect estimates for trial duration in seconds for Experiment 2.**

| | Estimate | Std. Error | *t* value |
|---|---|---|---|
| Intercept | 10.22 | 0.04 | 235.57 |
| Task Type | -0.60 | 0.03 | -17.39 |

unlinked words when reading for comprehension. Instead, we found the frequency effect occurred only for linked words during both reading for comprehension and skim reading. This suggests that when readers are able to navigate, their reading behaviour is more similar to skim reading of static webpages, even when reading for comprehension. When readers have the additional task of navigating through their reading environment, the importance of links is increased, causing them to be more fully processed than unlinked words. The lack of frequency effect for unlinked words regardless of task, combined with relatively low comprehension scores (compared to typical eye movement reading studies) suggests that the additional task of navigating is causing readers to enact a further speed-accuracy trade-off. As a result, readers base their reading behaviour around links, and are not fully lexically processing unlinked words due to their relative lack of importance to the reading task. It is also possible that high frequency words required less evaluation than low frequency words, as they denote more general, easier to understand concepts, hence the manifestation of these effects in later reading measures.

## General discussion

Across Experiment 1 and 2, the effects of skimming, word frequency (as an index of lexical processing) and linked words were investigated. In Experiment 1, participants read static Wikipedia pages on a screen, while Experiment 2 additionally introduced the ability to click and navigate through the webpages in a manner similar to typical Web browsing. By conducting these two experiments together we could dissociate the impact of hyperlinks on skim reading because hyperlinks are salient words, from the fact that hyperlinks additionally also serve as links for navigating on the Web. We provide strong findings for the impact of hyperlinks on skim reading and for the impact of navigating the text.

Across Experiment 1 and 2, we replicated the effects of skim reading previously observed by Just and Carpenter [6]. Specifically, we observed shorter fixations, increased skipping rates and reduced comprehension accuracy when skim reading condition compared to the reading for comprehension task. We also observed robust word frequency effects, whereby low frequency words had longer fixation times compared to high frequency words, [6–8]. Importantly, we also provide a replication of effects observed in Fitzsimmons et al. [30], whereby the presence of hyperlinks did not have a strong negative impact on reading. When reading for comprehension in a static Web environment (Experiment 1), there was no effect of whether the word was a hyperlink or not on early processing, with effects limited to increased re-reading of the low frequency, hyperlinked words (Experiment 2, Fig 6), possibly due to the need to evaluate these words as links.

The staggered introduction of skim reading, and navigation allowed for the dissociation of their influences on reading, resulting in several novel findings. When skim reading, we found a reduction in lexical processing for unlinked words (Experiment 1 and 2), evidenced by the lack of frequency effect for unlinked words, in contrast to Just and Carpenter [6] who still found a frequency effect during skim reading. This indicates that the additional context of reading webpages causes a reduction in lexical processing for information deemed less important, when skim reading. Conversely, hyperlinks still showed a frequency effect when skim reading static webpages, indicating readers deemed them of importance, resulting in deeper lexical processing. The introduction of an interactive, Web-like environment in Experiment 2 allows investigation of the actual impact of links and reading on the Web. The unique addition of this task element showed readers did not show a frequency effect for unlinked words, regardless of task, whereas the frequency effect was observed for linked words. To be clear, task type did not modulate whether there was a frequency effect for unlinked words. This

indicated that the additional ability to navigate through text caused readers to not fully lexically process unlinked words, even when reading for comprehension. This further supports the notion that reading on the Web is more likely to involve reading patterns akin to skim reading [3, 4] and suggests lexical representations of text are less detailed than those during typical reading.

Clearly, the Web environment places different demands on readers, resulting in adaptive strategies, in order to gain as much information as possible from the text, while keeping reading time short. The adaptive strategy appears to involve readers not fully processing unlinked words when skim reading static webpages, or when reading navigable webpages (regardless of task). This adaptive strategy could be likened to information foraging theory. Pirolli and Card explain information foraging theory using a metaphor of a bird foraging for berries in a patch of bushes [45]. The bird must decide how long to spend on one patch of bushes before spending time moving onto a new patch to find berries. The most efficient time to leave for a new patch is when the expected future gains from foraging in the current patch decreases to such a level that it is better to spend time moving to a new patch. In terms of foraging for information while reading, it is assumed that readers are sensitive to their 'information gain' (how much useful information they are obtaining over time) and they can use this as a basis for what to read and when to stop reading. If information gain drops below a threshold they will stop reading that particular piece of text and move on to a new section of text where they might gain more information in the same amount of time. The new 'patch' could represent a new line, new paragraph or new webpage.

Reader and Payne suggested that this information foraging approach of satisficing can be applied to skim reading if we assume that the 'patches' are patches of text or paragraphs, and the reader has a threshold for their information gain influenced by the amount of time they have to read the text [18]. For example, if the reader has a short amount of time to read the text, they will have a lower threshold for information gain. If the reader is not receiving enough information from a patch they will want to realise this quickly because of the limited amount of time. Therefore, they will want to move on to a patch that has a higher information gain as soon as possible in order to make the most efficient use of the limited time. The readers will focus on the most important information patches and leave the patches with less important information as soon as they can in order to spend their time in the most effective and efficient way possible. It could be that when skim reading or reading with the intent to navigate to a new page, unlinked words do not represent enough interesting information to be fully processed, causing readers to quickly move onto other regions of the text, or a new webpage. Further research is required to investigate in more detail how hyperlinks affect readers' assessments of importance of information.

It is notable, however, that we found influences of navigation on lexical processing in a task where navigation did not affect the informational goals of the reader. Experiment 2 provides a baseline for how readers lexically process text when navigating hypertext (for both skim reading and reading for comprehension). Further research should investigate whether these differences in reading are modulated by the importance of the information to comprehending the webpage, and how this is further affected by other informational goals. This is of particular importance given the role of informational goals in signalling theory for text (e.g. the SARA model [42]). Clearly, not all information is equal in terms of understanding the semantic content of passages, and this assessment will be made to ensure efficiency during skim reading. While the studies reported here investigate the use of hyperlinks to indicate importance, this can also be further investigated through the semantic content of passages. Further research is required to investigate how these higher-level semantic factors affect reading on the Web and the level of lexical processing that occurs within this task, beyond the baseline provided here.

Comprehension rates were much lower in Experiment 2, compared to Experiment 1 and other typical eye movement studies (Experiment 1, Reading for comprehension: 91%; Skim reading: 86%; Experiment 2 Reading for comprehension: 66%; Skim reading: 60%). However, the stimuli in Experiment 2 differ from Experiment 1, due to the large number of Wikipedia articles required to make a fully functioning Web environment (i.e. one with many links to many different pages, resulting in 843 Wikipedia articles in total for Experiment 2, compared to 40 in Experiment 1). The questioning rate also differed (Experiment 1: 4 questions per article; Experiment 2: 1 question for 45% of trials). Finally, participants in Experiment 2 saw different questions based on the different webpages they clicked on. Due to these content and task differences, it is difficult to compare the comprehension scores for these two studies. Previous research has suggested that evaluating hyperlinks and navigating a path through them substantially increases readers' cognitive load [24, 26, 27]). Our comprehension question data are compatible with this idea but as we already indicated the comparison between the accuracy on the comprehension data across the two experiments is problematic given content and task differences. As such future research should further explore whether increased cognitive load does indeed cause any cost to higher level comprehension and discourse representation.

In summary, we found that being able to navigate had a substantial effect on how readers process text in a hypertext environment. By comparing the findings from both experiments, we explored the impact navigating and clicking has on factors such as the task the reader is engaged in (reading for comprehension or skim reading), whether a target word is linked or not and the frequency of the target word. We observed that the reader places importance on linked target words and spent more time on them, especially the low frequency, linked target words. Additionally, we found that task demands lead to unlinked words receiving relatively little lexical processing as is evidenced by the lack of a frequency effect on them, whereas linked words are more deeply processed at a lexical level.

## Author Contributions

**Conceptualization:** Gemma Fitzsimmons, Lewis T. Jayes, Mark J. Weal, Denis Drieghe.

**Data curation:** Gemma Fitzsimmons, Lewis T. Jayes, Mark J. Weal, Denis Drieghe.

**Formal analysis:** Gemma Fitzsimmons, Lewis T. Jayes, Mark J. Weal, Denis Drieghe.

**Funding acquisition:** Gemma Fitzsimmons, Lewis T. Jayes, Mark J. Weal, Denis Drieghe.

**Investigation:** Gemma Fitzsimmons, Lewis T. Jayes, Mark J. Weal, Denis Drieghe.

**Methodology:** Gemma Fitzsimmons, Lewis T. Jayes, Mark J. Weal, Denis Drieghe.

**Project administration:** Gemma Fitzsimmons, Lewis T. Jayes, Mark J. Weal, Denis Drieghe.

**Resources:** Gemma Fitzsimmons, Lewis T. Jayes, Mark J. Weal, Denis Drieghe.

**Software:** Gemma Fitzsimmons, Lewis T. Jayes, Mark J. Weal, Denis Drieghe.

**Supervision:** Gemma Fitzsimmons, Lewis T. Jayes, Mark J. Weal, Denis Drieghe.

**Validation:** Gemma Fitzsimmons, Lewis T. Jayes, Mark J. Weal, Denis Drieghe.

**Visualization:** Gemma Fitzsimmons, Lewis T. Jayes, Mark J. Weal, Denis Drieghe.

**Writing – original draft:** Gemma Fitzsimmons, Lewis T. Jayes, Mark J. Weal, Denis Drieghe.

**Writing – review & editing:** Gemma Fitzsimmons, Lewis T. Jayes, Mark J. Weal, Denis Drieghe.

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
