## [Decision Letter · Decision Letter 0]

13 May 2020

PONE-D-20-06861

The Impact of Skim Reading and Navigation when Reading Hyperlinks on the Web

PLOS ONE

Dear Dear Dr. Fitzsimmons,

Thank you for submitting your manuscript to PLOS ONE. After careful consideration, we feel that it has merit but does not fully meet PLOS ONE’s publication criteria as it currently stands. Therefore, we invite you to submit a revised version of the manuscript that addresses the points raised during the review process.

As you will see below, the reviewers feel that your work will make an important contribution to the empirical literature; however, they raised a number of concerns regarding the methodology, statistical analyses (including accurate presentation/interpretation thereof), and overall clarity of your paper that preclude it from publication at this point. I urge you to pay close attention to their feedback; however, I do not agree with Reviewer 3's recommendation of including ANOVAs rather than LMMs in your paper (although you are welcome to present the the findings ANOVA-style in your rebuttal letter). 

We would appreciate receiving your revised manuscript by July 13, 2020. To enhance the reproducibility of your results, we recommend that if applicable you deposit your laboratory protocols in protocols.io, where a protocol can be assigned its own identifier (DOI) such that it can be cited independently in the future. For instructions see: http://journals.plos.org/plosone/s/submission-guidelines#loc-laboratory-protocols

We look forward to receiving your revised manuscript.

Kind regards,

Veronica Whitford, Ph.D.

Academic Editor

PLOS ONE

2. In your Data Availability statement, you have not specified where the minimal data set underlying the results described in your manuscript can be found; the information provided does not provide a result that can be found as a data set.

PLOS defines a study's minimal data set as the underlying data used to reach the conclusions drawn in the manuscript and any additional data required to replicate the reported study findings in their entirety. All PLOS journals require that the minimal data set be made fully available. For more information about our data policy, please see http://journals.plos.org/plosone/s/data-availability.

3. Please upload a copy of Figures 6 and 7, to which you refer in your text on pages 27, 29, 31 and 32. If either figure is no longer to be included as part of the submission please remove all reference to it within the text.

Reviewers' comments:

Reviewer's Responses to Questions

**Comments to the Author**

1. Is the manuscript technically sound, and do the data support the conclusions?

Reviewer #1: Partly

Reviewer #2: No

Reviewer #3: Partly

2. Has the statistical analysis been performed appropriately and rigorously? 

Reviewer #1: I Don't Know

Reviewer #2: Yes

Reviewer #3: Yes

3. Have the authors made all data underlying the findings in their manuscript fully available?

Reviewer #1: No

Reviewer #2: Yes

Reviewer #3: Yes

4. Is the manuscript presented in an intelligible fashion and written in standard English?

Reviewer #1: Yes

Reviewer #2: Yes

Reviewer #3: Yes

5. Review Comments to the Author

Reviewer #1: This manuscript describes two experiments examining the impact of hyperlinks on eye movement behaviour during skim reading and reading for comprehension. Experiment 1 uses static pages, Experiment 2 uses pages with navigable links.

I thought that the Introduction and Experiment 1 read nicely, and I have only minor comments on those. However, there seem to be some major errors in the results section of Experiment 2. The statistics reported in Table 4 don’t match up to the text in several important respects, and also seem odd in relation to the means reported in Table 3. For example, if we take go-past time, according to Table 4 the only significant interaction is between word type and task type. However, the text (lines 617-626) reports and investigates a significant interaction between word frequency and word type, and makes no mention of an interaction between word type and task type. When comparing the means in Table 3 to the statistics in Table 4, it would also seem surprising that what appears numerically to be an enormous interaction between task type and word type for go-past time (difference between word type conditions 1014ms when reading for comprehension and 179ms when skim reading) has a t value of only 2.09. Obviously I appreciate that the log transformation will affect things, but given the other issues this may also be an error. These discrepancies are also present for the other measures in this experiment. In light of these I can’t tell what the results actually were, so I’m not in a position to comment on any of the discussion.

Minor comments

It would be helpful to provide a brief description of hyperlinks and hypertext (I was unfamiliar with the latter term).

Line 74 – Liu rather than Lui

Line 170 – I would suggest removing “where” as it creates a garden path

The example stimulus figures seem to be missing, and the numbers of the rest don’t match up with the text.

In both of your experiments you have the reading for comprehension block before the skim reading block. I agree that this is the best approach to take, but it may be worth considering what impact this may have had on your results.

In the tables it would make things easier if you had some lines or spaces separating the interactions – at the moment it’s difficult to work out which number corresponds to which interaction.

Line 382 – As you state that comprehension accuracy was lower for skim reading than reading for comprehension, it would be helpful to provide some statistics for this in the results section.

Lines 385-391 – It would be better to point out earlier in this section that this study only looked at reading for comprehension.

In Experiment 2 I presume there is necessarily variability in how many data points were available per participant in each condition. Therefore it would be good to have some sense of how much data there was in total and how this was distributed across conditions.

Lines 561-563 – Just for gaze duration? It is reported for all the other measures.

It’s very difficult to see the figures. I appreciate that they always end up in low resolution for reviews, but it would help if the layout was changed a bit (e.g. for the figures where you have three graphs, they could be displayed larger if placed on top of each other rather than side by side).

Reviewer #2: Thanks for the opportunity to assess the manuscript “The Impact of Skim Reading and Navigation when Reading Hyperlinks on the Web”. This is a well though, nicely controlled set of studies which aim to identify how hypertexts, and particularly hyperlinks, are processed by means of eye-tracking indicators. Unfortunately, the strong emphasis on experimental controlled materials came to the expense of limiting the construct validity of the task. As a result, the studies resemble more a traditional reading comprehension tasks than hypertext ones. Let me elaborate on this relevant point in the following.

The studies used only word level indicators of eye-movements to describe what they define as “reading behavior”. This approach is representative of the vast majority of psycholinguistic studies of reading, such as the ones critically reviewed in the introduction. As the authors claimed, such approach allowed them to build a “well-controlled” study, unique in this area of hypertext reading. In this regard, the methods used are strong and accordingly the results based on the eye-movements indicators are reliable. However, the use of only word level indicators severely limits the scope of the studies, for at least two reasons. Before elaborating on those, I should remark that there are alternative eye-movements indicators beyond word level ones, which allow to capture strategic reading behaviors, such as the ones we may expect to arose during hypertext reading. Among those, there are paragraph level measures, sequence indicators… Those have been used both in text reading research (e.g. Hyönä et al., 2000, JEdP) as well as in hypertext reading (e.g. Dugan & Paine, 2009, JEP; Salmerón et al., 2017, JCAL).

Regarding the problems associated to deciding to focus only on word level measures, the first one is that this methodological paradigm forces researchers to make strong design decisions to adjust the reading materials to maximize the experimental control. This is the same critique introduced in the manuscript when describing traditional sentence-long texts research literature on reading. Thus, a “well-controlled” study can become a reliable study but with poor construct validity. The most obvious limitation here is the claim that Experiment 1 was about hypertext reading, while no navigation was allowed. To me, such study corresponded clearly to the reading literature of text signaling devices (e.g. work by Lorch and Lemarié), but not hypertext, where navigation must be present. In Experiment 2, the limits of this approach were evident when authors decided that the task will be finished once participants have read ten pages. From the perspective of the participant as reading/student, I struggle to understand what would mean to me to “read a hypertext until you have reached 10 pages” (I assume this was told to the participants, although it was not clear to me what was exactly told).

In the same line, in Experiment 2 pages were constructed to be independent from each other, if only because the comprehension questions only targeted information stated in a single page, and the participants are not given any particular reading goal emphasizing the need to strategically navigate the hypertext. This is a clever decision when you want to focus on word (or page for the matter) as an item, but it nevertheless makes navigation irrelevant in this context. Without any goal to guide her navigation, what is the participant meant to do? The two theoretical models discussed in the conclusion, the information foraging theory and the satisfying heuristic, start by providing participants with a specific goal (e.g. to find information, to write a summary, etc). Navigation takes place to efficiently achieve such goal.

On a different note, I have several concerns regarding the type of comprehension questions used in both studies:

-In experiment 1, comprehension scores were really high, suggesting that there was a floor effect. This limits the chances to capture differences between conditions.

-In both studies, questions only asked information presented in a single page. This represents a rather limited view of text comprehension, as it ignores integration processes that take place when students relate information across pages.

-In both studies, if I got it right, comprehension questions were provided immediately after leaving a page, instead of being provided at the end of the reading session. Such combination emphasized the idea that participants had to focus on the single pages, and that there was no need to strategically navigate across the pages. Why not just giving the comprehension questions after the reading session?

-In experiment 2 I find it hard to understand how comprehension could be assessed, and compared across participants, given that participants could read a different set of materials in each condition, and questions were aligned to specific pages. Is there any evidence that the difficulty of the comprehension questions was similar across pages?

All in all, I have confronted feelings on the manuscript. On the one hand, I really value the effort to build well controlled studies in a complex scenario such as hypertext reading. The approach taken in these studies is not only excellent but also necessary in an area in need for well controlled studies. But in this particular attempt, I am afraid that authors have gone too far by overvaluing experimental control over construct validity.

Minor point

Table 1. (190 or (19)?

Reviewer #3: In the manuscript entitled "The Impact of Skim Reading and Navigation when Reading Hyperlinks on the Web" the authors report two experiments using eye-tracking measures to study effects of hyperlinks on reading behavior. The authors describe a complete within-subjects design, in which they experimentally manipulated three factors: reading for comprehension or skim reading (i.e., task), target words marked as hyperlinks or not, and target words of high frequency or of low frequency. In Exp. 1 participants had to read only, in Exp. 2 participants could navigate through an entire hypertext-environment by clicking on the words marked as hyperlinks. The authors observed in Exp. 1 a word frequency effect (i.e., a difference between words of low and high frequency) to be present only during skim reading for words marked as hyperlinks. This outcome was interpreted as showing that during skim reading only hyperlinked-words were fully lexically processed. In contrast, in Exp. 2 when the hyperlinked words were clickable, the authors observed under both task conditions that only words marked as hyperlinks were fully lexically processed.

The manuscript is generally well written and adds some additional insights in the effects of hyperlinks on reading, strongly building upon previous research conducted by the authors.

However, I do have some major concerns the authors might thoroughly address before publication. Most of these concerns equally address both experiments, although I might only refer to specific examples of the first or the second experiment.

1) I missed a more detailed description of the actual task materials used respectively had some problems in understanding the design of the task materials based on the description provided by the authors (this holds true for Exp. 1 and 2).

For Exp. 1&2: What were the topics of the 40 articles and how long were the articles? Based on which ratio were the target words in the sentences selected? Were the target words at a specific position within the sentence or equally distributed across the sentences? How long were the sentences? Were the target words of a specific category? Were the sentences of a specific topic?

If I understand it correctly, the experimental manipulations in Exp. 1 were done in specific sentences that were then inserted into pre-existing Wikipedia-articles. How was the topical congruency between the inserted sentences and the rest of the article ensured? Did this procedure result in realistic Wikipedia-like articles?

As the authors note, pre-existing hyperlinks in the articles were retained (the authors might specify the number of those 'other' hyperlinks). If I understand that correctly, there was an unbalance in the total number of hyperlinks in the articles based on whether the articles were used for the hyperlink-condition or the no-hyperlink-condition, that is in the hyperlink-condition there were more hyperlinks in the article as compared to the no-hyperlink condition. Potentially, the presence of additional hyperlinks in the no-hyperlink-condition could have severely influenced the processing of the non-hyperlinked target word, that is, at least partly explain the reduced lexical processing thereof during skim reading as participants might have predominantly focused on the remaining hyperlinks.

Besides, Figure 1 and Figure 5 that should show examples of the task materials are missing, hindering a correct understanding and discussion of the materials.

2) I suspect (but maybe I am wrong) in Exp. 1 the manipulation of hyperlink present/not present and word frequency was done article-wise. If this interpretation is correct than that’s an additional difference to Exp. 2 that should be discussed as in Exp. 2 both hyperlinked and non-hyperlinked target words had to be present within the same article to allow for navigation. It would be helpful if the authors thoroughly revised these sections of the manuscript. Or to put it in other words: did each article contain each manipulation (i.e. equal amounts of low/high frequency words, hyperlinks/no hyperlinks) or were the conditions manipulated per article (i.e., e.g., 5 articles skim reading and low frequency target words and no hyperlinks)? Were the experimental manipulations bound to a specific article or were they equally distributed across articles?

3) The fixed order of first, reading for comprehension, and second, skim reading in both experiments is problematic, as it confounds the manipulation of reading task with potential sequential effects (like tiredness, boredom, or the fact that participants generally might getting used to the overall task). Therefore, it is questionable whether the observed effects on skim reading could really be attributed to skim reading or might be (at least partly) due to confounding factors. The authors clearly should state this as a limitation of their study that has to be discussed more thoroughly in the general discussion section. Maybe in an additional analysis, the authors could split the reading for comprehension phase into two parts and see whether in the second part the reading behavior has already become more skim-reading like. If this is not the case, this outcome might be used as an argument why the fixed sequence might not be problematic. Besides, the argumentation of the authors for the fixed order could be also reversed. Potentially, if the participants start reading for comprehension at the beginning they might continue to do so throughout the entire experiment (especially due to the repeated knowledge tests). The authors should report the entire reading times for the two tasks in Exp. 1 and might analyze reading times for differences between skim reading and reading for comprehension.

4) Were the knowledge tests specifically related to the target words/sentences or rather to the rest of the article?

5) What exactly were the task instructions for the two tasks? Were the participants informed about the following knowledge questions? How (i.e., did they get a global task of remembering everything or specific aspects of the texts only?)?

6) I would be interested to hear some arguments of the authors why they decided to use linear mixed models for their statistical analyses. Based on the clear 3-factorial task design and the research question I in a way would have expected classical repeated-measures ANOVAs. What is the addon of using LMM in this scenario? Besides, I would be curious whether repeated-measures ANOVAs would show up identical results.

7) I miss a thorough introduction and motivation of the eye-tracking measures used in the two studies (also backed up with more literature). For example, in the introduction the authors state skim reading is characterized by fewer fixations on words, but this measure is not used by the authors to disentangle skim reading from thorough reading. Reporting such additional, established eye-tracking measures might be valuable as a manipulation-check of the reading tasks. Besides, for me it did not become fully clear why the authors did not use the same measures in both experiments and why they only focus on measures with respect to the target words. Especially in the non-hyperlink condition, the target words were opaque to the participants, so it is kind of trivial that participants fixated hyperlinked target words differently than the other words (this in a way relates to my question, what was specific about the target words). As the authors state, they were interested in how hyperlinks and hyperlink-selection affects reading behavior I would have expected to hear more about more global eye-tracking measures on the sentence or text level (like e.g., total number of fixations).

Is the measure 'go-past time' equivalent to the dwell time?

8) The transition from Exp. 1 to Exp. 2 might be kind of optimized. Based on the discussion of Exp. 1 I would have expected the story would go to look at the effects of highlighted words in general (which the authors already addressed in previous research, so this research might be referenced here). In a way, I am missing a more thorough motivation for conducting Exp. 2. Besides, as the authors started the manuscript by describing classical reading research and the artificial task materials used in these studies, one might state that the task materials the authors used in the current study still seem kind of artificially (which is legitime in doing good research, yet, the argument of ecological validity implicitly stated at the beginning of the manuscript might still only partly be redeemed. For example, in Exp. 2 the single Wikipedia-articles seem to consist of only some sentences (and in addition no other elements, like pictures), therefore the question arises how realistic the material was actually percieved by the participants.

9) The interpretation that target words were differently lexically processed seems to be a quite strong conclusion given that no direct measure of (or probe for) lexical processing is provided.

10) Were the comprehension scores in Exp. 2 significantly different for the two reading tasks?

Some minor comments:

• There are some typos (e.g., Table 1 a 'zero' instead of a closing bracket) and some grammatical inconsistencies (e.g., sometimes use of simple present instead of past tense when describing the study).

• The result section of Exp. 2 might be more closely structured as that of Exp1 (especially with respect to sub-headings).

• Exp. 1 states that participants did not take part in Exp. 2. However, a comparable information is missing in Exp. 2. Does this mean Exp. 2 was actually conducted before Exp. 1?

• Exp. 2 how many target words (i.e., trials) on total get into the statistical analysis (i.e., how many potential target words were excluded from analysis due to the criterion stated l. 521? Could this result in an unbalanced amount of target words for the hyperlinked/non-hyperlinked task conditions?)?

• Exp. 2: How many hyperlinks were actually clicked? Was there a different clicking-pattern between the task conditions (i.e., skim reading or reading for comprehension)?

• Figures: Maybe the figures could be replotted in a more condensed way, e.g. with the data for the two reading tasks in the same plot (e.g., using color-coding). This might visually help in comparing the outcomes of the experimental conditions and would reduce the (rather high) total number of separate plots.

6. PLOS authors have the option to publish the peer review history of their article (what does this mean?). If published, this will include your full peer review and any attached files.

Reviewer #1: No

Reviewer #2: No

Reviewer #3: No

---

## [Author Response · Author response to Decision Letter 0]

12 Jun 2020

Included in 'Response to Reviewers' document. We hope that you will find our replies and revisions to satisfactorily address your and the reviewers’ concerns.

---

## [Decision Letter · Decision Letter 1]

24 Jul 2020

PONE-D-20-06861R1

The Impact of Skim Reading and Navigation when Reading Hyperlinks on the Web

PLOS ONE

Dear Dr. Fitzsimmons,

Thank you for submitting your manuscript to PLOS ONE. The reviewers were generally pleased with the changes implemented in your revised manuscript; however, there are some relatively minor residual concerns that need to be addressed before it is accepted for publication. These include issues with data reporting and clarity of the Introduction. Could you please re-submit your revised manuscript by September 1, 2020? If you will need more time than this to complete your revisions, please reply to this message or contact the journal office at plosone@plos.org. Please include the following items when submitting your revised manuscript:

We look forward to receiving your revised manuscript.

Kind regards,

Veronica Whitford, Ph.D.

Academic Editor

PLOS ONE

**Comments to the Author**

1. If the authors have adequately addressed your comments raised in a previous round of review and you feel that this manuscript is now acceptable for publication, you may indicate that here to bypass the “Comments to the Author” section, enter your conflict of interest statement in the “Confidential to Editor” section, and submit your "Accept" recommendation.

Reviewer #1: (No Response)

Reviewer #3: (No Response)

2. Is the manuscript technically sound, and do the data support the conclusions?

Reviewer #1: Yes

Reviewer #3: Yes

3. Has the statistical analysis been performed appropriately and rigorously? 

Reviewer #1: Yes

Reviewer #3: I Don't Know

4. Have the authors made all data underlying the findings in their manuscript fully available?

Reviewer #1: Yes

Reviewer #3: Yes

5. Is the manuscript presented in an intelligible fashion and written in standard English?

Reviewer #1: Yes

Reviewer #3: Yes

6. Review Comments to the Author

Reviewer #1: This manuscript reports two experiments examining the impact of hyperlinks on eye movement behaviour during reading. I was reviewer 1 previously.

My main concern with the previous manuscript was that there appeared to be some errors in the reporting of the results. I am satisfied that this was simply a labelling error, and everything seems to now match up. I have only some minor comments about this updated manuscript:

Abstract lines 26-28 – This is a little ambiguous – it could be interpreted as that readers only lexically process linked words when skim reading and not when reading for comprehension. There is a similar issue in line 434.

Lines 359-361 – You report skipping probabilities here, but these don’t seem to match with what is reported in table 1.

I presume the data in Figure 3 has been log-transformed (as it has for the linear mixed effects models). This should be mentioned (perhaps on the label for the Y axis) as this is fairly unusual.

Reviewer #3: In the revised version of the manuscript entitled "The Impact of Skim Reading and Navigation when Reading Hyperlinks on the Web" the authors addressed most of my previous concerns satisfactorily.

Yet I still do have some (mostly minor) concerns.

The authors now provide a more detailed description of the task materials. Still I do think an example of the task materials used should be shown directly in the manuscript as a figure. It would be nice to directly highlight in this exemplary task material the additionally inserted sentences and hyperlinks (i.e., making the experimental manipulation directly visible). Note that I am rather asking for a schematic depiction of the task materials showing the experimental manipulation at a glance, not necessarily the real materials with the Wikipedia logo. The real task material still could be provided via a hyperlink, yet I am kind of skeptical whether a link to Google drive is a very sustainable solution (i.e., can the authors ensure that the link is working in five or ten years?). Generally, I think for readers it is preferable to have all relevant information on a study in the manuscript at hand and not only via references or links to other sources.

As the authors now clearly limited the scope of their research question to the lexical processing of hyperlinks at the word level, I suggest also revising the abstract accordingly. In its current form, the initial sentences still raise the expectation that the authors' study deal with the effects of hyperlinks on text comprehension and reading behavior. Yet, the authors specifically only focus on the word level (and only on specific hyperlinked words). This is a valid approach, but it should be made very clear early in time. Also, with respect to the last sentence of the abstract one might wonder whether this conclusion is not too broad (and strong) given the specific focus of the experiments.

The authors might also think about revising the beginning of the introduction. Starting with differentiating reading online text versus reading in reading experiments is kind of misleading as the authors' study does not a direct comparison of these two reading tasks. Also, the different reading strategies are induced via instruction, that is, skim reading could also be instructed in classical reading experiments. Besides, offline reading might be considered also to be very different from reading in typical reading experiments. As noted for the abstract, I would suggest being clear about the specific focus of the authors' study early in time to avoid raising false expectations among readers.

l. 57: A short definition of skim reading might be helpful for the general reader here.

The use of the general term 'reading behavior' (e.g., used l. 191) might be considered as being too broad for the specific focus of the authors' study. The authors might clarify (i.e., define) early in time what is meant by reading behavior in the context of the current study.

l.299: The authors might provide a reference for their outlier definitions of fixation durations (i.e., why merging fixations shorter 80 ms, why not only those shorter 50 ms; why removing fixations longer 800 ms).

7. PLOS authors have the option to publish the peer review history of their article (what does this mean?). If published, this will include your full peer review and any attached files.

Reviewer #1: No

Reviewer #3: No

---

## [Author Response · Author response to Decision Letter 1]

28 Jul 2020

Thank you for submitting your manuscript to PLOS ONE. The reviewers were generally pleased with the changes implemented in your revised manuscript; however, there are some relatively minor residual concerns that need to be addressed before it is accepted for publication. These include issues with data reporting and clarity of the Introduction. 

We have now made the suggested changes, particularly regarding data reporting and the introduction.

Reviewer #1: This manuscript reports two experiments examining the impact of hyperlinks on eye movement behaviour during reading. I was reviewer 1 previously.

My main concern with the previous manuscript was that there appeared to be some errors in the reporting of the results. I am satisfied that this was simply a labelling error, and everything seems to now match up. I have only some minor comments about this updated manuscript:

Abstract lines 26-28 – This is a little ambiguous – it could be interpreted as that readers only lexically process linked words when skim reading and not when reading for comprehension. There is a similar issue in line 434.

We have altered the wording of these two sentences to remove the ambiguity over the findings between skim reading and reading for comprehension. 

Lines 359-361 – You report skipping probabilities here, but these don’t seem to match with what is reported in table 1.

We have removed this incorrect information from the manuscript, the reader is now referred to table one, where it can be seen that skipping rates are significantly higher for high frequency words than low frequency. 

I presume the data in Figure 3 has been log-transformed (as it has for the linear mixed effects models). This should be mentioned (perhaps on the label for the Y axis) as this is fairly unusual.

The go-past time measures were log transformed in this experiment, and Figure 3 displays the results from this LMM. We now also mention that these data are log transformed in the caption for this figure to make this clear. 

Reviewer #3: In the revised version of the manuscript entitled "The Impact of Skim Reading and Navigation when Reading Hyperlinks on the Web" the authors addressed most of my previous concerns satisfactorily.

Yet I still do have some (mostly minor) concerns.

The authors now provide a more detailed description of the task materials. Still I do think an example of the task materials used should be shown directly in the manuscript as a figure. It would be nice to directly highlight in this exemplary task material the additionally inserted sentences and hyperlinks (i.e., making the experimental manipulation directly visible). Note that I am rather asking for a schematic depiction of the task materials showing the experimental manipulation at a glance, not necessarily the real materials with the Wikipedia logo. The real task material still could be provided via a hyperlink, yet I am kind of skeptical whether a link to Google drive is a very sustainable solution (i.e., can the authors ensure that the link is working in five or ten years?). Generally, I think for readers it is preferable to have all relevant information on a study in the manuscript at hand and not only via references or links to other sources.

We have added an additional schematic figure to convey the stimuli, as well as the high/low frequency and linked/unlinked manipulations (figure 1). 

As the authors now clearly limited the scope of their research question to the lexical processing of hyperlinks at the word level, I suggest also revising the abstract accordingly. In its current form, the initial sentences still raise the expectation that the authors' study deal with the effects of hyperlinks on text comprehension and reading behavior. Yet, the authors specifically only focus on the word level (and only on specific hyperlinked words). This is a valid approach, but it should be made very clear early in time. Also, with respect to the last sentence of the abstract one might wonder whether this conclusion is not too broad (and strong) given the specific focus of the experiments.

We have altered the wording of the of the abstract (particularly the first and last sentence) to show the reduced scope of our findings to lexical processing of single words. 

The authors might also think about revising the beginning of the introduction. Starting with differentiating reading online text versus reading in reading experiments is kind of misleading as the authors' study does not a direct comparison of these two reading tasks. Also, the different reading strategies are induced via instruction, that is, skim reading could also be instructed in classical reading experiments. Besides, offlinen reading might be considered also to be very different from reading in typical reading experiments. As noted for the abstract, I would suggest being clear about the specific focus of the authors' study early in time to avoid raising false expectations among readers.

We have amended our discussion of the focus of the study to be more specific, to avoid any ‘false expectations’. 

l. 57: A short definition of skim reading might be helpful for the general reader here.

A short definition of skim reading is now provided when it is first mentioned in the Introduction.

The use of the general term 'reading behavior' (e.g., used l. 191) might be considered as being too broad for the specific focus of the authors' study. The authors might clarify (i.e., define) early in time what is meant by reading behavior in the context of the current study.

While we maintain the use of reading behaviour, as that is what our participants were engaged in by the skim/comprehension manipulation, we have included more references throughout to its effect on lexical processing, as it is what our dependent variables represent. We have made minor amendments to make this distinction clearer within the text. 

l.299: The authors might provide a reference for their outlier definitions of fixation durations (i.e., why merging fixations shorter 80 ms, why not only those shorter 50 ms; why removing fixations longer 800 ms).

We now provide references regarding data cleaning to evidence our procedures for this were the standard with eye movement and reading studies.

---

## [Decision Letter · Decision Letter 2]

1 Sep 2020

The Impact of Skim Reading and Navigation when Reading Hyperlinks on the Web

PONE-D-20-06861R2

Dear Dr. Fitzsimmons,

We’re pleased to inform you that your manuscript has been judged scientifically suitable for publication and will be formally accepted for publication once it meets all outstanding technical requirements.

Kind regards,

Veronica Whitford, Ph.D.

Academic Editor

PLOS ONE

Reviewers' comments:

Reviewer's Responses to Questions

**Comments to the Author**

1. If the authors have adequately addressed your comments raised in a previous round of review and you feel that this manuscript is now acceptable for publication, you may indicate that here to bypass the “Comments to the Author” section, enter your conflict of interest statement in the “Confidential to Editor” section, and submit your "Accept" recommendation.

Reviewer #1: All comments have been addressed

Reviewer #3: All comments have been addressed

2. Is the manuscript technically sound, and do the data support the conclusions?

Reviewer #1: (No Response)

Reviewer #3: (No Response)

3. Has the statistical analysis been performed appropriately and rigorously? 

Reviewer #1: (No Response)

Reviewer #3: (No Response)

4. Have the authors made all data underlying the findings in their manuscript fully available?

Reviewer #1: (No Response)

Reviewer #3: (No Response)

5. Is the manuscript presented in an intelligible fashion and written in standard English?

Reviewer #1: (No Response)

Reviewer #3: (No Response)

6. Review Comments to the Author

Reviewer #1: (No Response)

Reviewer #3: (No Response)

7. PLOS authors have the option to publish the peer review history of their article (what does this mean?). If published, this will include your full peer review and any attached files.

Reviewer #1: No

Reviewer #3: No

---

## [Editor Report · Acceptance letter]

4 Sep 2020

PONE-D-20-06861R2 

The Impact of Skim Reading and Navigation when Reading Hyperlinks on the Web 

Dear Dr. Fitzsimmons:

I'm pleased to inform you that your manuscript has been deemed suitable for publication in PLOS ONE. Congratulations! Your manuscript is now with our production department. 

Kind regards, 

on behalf of

Dr. Veronica Whitford 

Academic Editor

PLOS ONE